# Development of an Exoskeleton Platform of the Finger for Objective Patient Monitoring in Rehabilitation

**DOI:** 10.3390/s22134804

**Published:** 2022-06-25

**Authors:** Nikolas Jakob Wilhelm, Sami Haddadin, Jan Josef Lang, Carina Micheler, Florian Hinterwimmer, Anselm Reiners, Rainer Burgkart, Claudio Glowalla

**Affiliations:** 1Department of Orthopedics and Sports Orthopedics, Klinikum rechts der Isar, School of Medicine, 80333 Munich, Germany; jan.lang@tum.de (J.J.L.); carina.micheler@tum.de (C.M.); florian.hinterwimmer@tum.de (F.H.); rainer.burgkart@tum.de (R.B.); claudio.glowalla@bgu-murnau.de (C.G.); 2Munich Institute of Robotics and Machine Intelligence, Department of Electrical and Computer Engineering, Technical University of Munich, 80333 Munich, Germany; sami.haddadin@tum.de; 3Klinik für Frührehabilitation und Physikalische Medizin, Zentrum für Orthopädie, Unfallchirurgie und Sportmedizin, München Klinik Bogenhausen, 81925 Munich, Germany; anselm.reiners@muenchen-klinik.de; 4Department of Trauma and Orthopedic Surgery, Berufsgenossenschaftliche Unfallklinik Murnau, 82418 Murnau, Germany

**Keywords:** exoskeleton, hand, CRPS, rehabilitation, mobile application, clinical study

## Abstract

This paper presents the application of an adaptive exoskeleton for finger rehabilitation. The system consists of a force-controlled exoskeleton of the finger and wireless coupling to a mobile application for the rehabilitation of complex regional pain syndrome (CRPS) patients. The exoskeleton has sensors for motion detection and force control as well as a wireless communication module. The proposed mobile application allows to interactively control the exoskeleton, store collected patient-specific data, and motivate the patient for therapy by means of gamification. The exoskeleton was applied to three CRPS patients over a period of six weeks. We present the design of the exoskeleton, the mobile application with its game content, and the results of the performed preliminary patient study. The exoskeleton system showed good applicability; recorded data can be used for objective therapy evaluation.

## 1. Introduction

Patients with impaired hand functioning may experience severe limitations in daily life [1,2,3,4,5,6]. Limited hand mobility can be caused by various factors, including neuromuscular diseases, injuries, restricted motor functions from strokes, or age-related limitations [3]. Further, the complex regional pain syndrome can affect hand functionality, as patients are hypersensitive to stimuli or touch and have limitations of movement, such as fist closure or tweezer grip [7].

The main therapeutic goals in CRPS are to maintain or improve hand mobility and functioning and to reduce pain. Physiotherapy and occupational therapy are two central pillars of conservative treatments, which aim to compensate for pathological movement patterns and prevent long-term damage, such as contractures due to insufficient use as a result of pain avoidance [8]. In this context, the effectiveness of physiotherapy individually adapted to the patient using mobilization and traction therapy has been proven. The focus is on repetitive pain-free movements and training of fine motor skills with and without resistance. The duration of physiotherapy or ergotherapy should be 20–30 min a day with 5 therapy sessions per week if possible [9,10].

The conventional method of manual treatment is time-consuming and often leads to unsatisfactory results, as therapy is often too short, requires an expert therapist, and has high financial burdens [11,12]. Further, as patient data are not actively collected, patient progress is only measured subjectively. In contrast, exoskeletons can provide accurate data to track the functional status of the patient’s hand, while not increasing the financial burden of the treatment over long periods or frequent applications. This enables patients to follow daily therapy (usually not applicable in manual therapy) and maintain therapy over a long period of time. However, the use of an exoskeleton system in therapy is still limited, as only a small subset makes it into clinical testing or practice due to the complexity and resulting in poor usability in a clinical context [5]. Numerous exoskeletons have been developed, often requiring a direct fit to the patient’s hand [13,14,15], or providing an adaptive actuation [6,16,17]. Often, these concepts do not provide sufficient sensor applications and lack sufficient force sensing or cannot distinguish between the movements of MCP, PIP, and DIP joints.

The approach by Dickmann et al. [18] extended the approach by Conti et al. [6] and provided sufficient sensing for index finger actuation, motion tracking, and force control. We extended this approach to be suitable for clinical use. A mobile control application was developed; live tracking of patient data and easy applicability are ensured, making it possible to simplify the use of the exoskeleton and provide the system as an extension of conventional therapy, objectifying the patient’s progress through the sensor data collected. In summary, we make the following contributions:We extended the exoskeleton by Dickmann et al. [18] to incorporate a Bluetooth remote control and connection to a newly developed mobile application.The application enables adaptive, individual control and is enhanced by two games to motivate patients for longer therapy sessions and record objective data.We performed a small patient study with three patients and tracked their longitudinal progress over six weeks; we present the results.We compare the results of the patient to the subject study from Dickmann et al. [18].

## 2. Materials and Methods

### 2.1. Mechatronic System

The mechatronic system extends the approach by Dickmann et al. [18] to include a Bluetooth control unit and improved housing and packaging of all electrical components. The electrical circuit is shown in Figure 1, including the individual electrical components and their application for the exoskeleton system.

The three potentiometers used to measure the angles of the exoskeleton kinematics are shown in grey. The finger length could be used to calculate the individual joint angles. The two force-sensing resistor (FSR) units determined the external force applied by the actuator and are displayed in blue [18]. The motor is powered by an external voltage source and receives a pulse width modulator (PWM) signal from the microcontroller for control. The newly integrated DSDTech HM10 Bluetooth module enables external Bluetooth communication and sends the signals from the three potentiometers and the force signal with 50 Hz to the mobile application. Conversely, it receives the commands for executing measurement protocols and actuator trajectories from the application.

### 2.2. Exoskeleton Framework

The architecture of the exoskeleton system is shown in Figure 2.

The basis of the developed concept is the exoskeleton by Dickmann et al. [18] for the index finger. It can be attached to any hand due to the adaptive approach. With the help of the sensors, all forces and torques in the finger joints and their positions can be determined [18]. The exoskeleton is shown in Figure 2a. With the help of the therapist, and in later steps independently, the exoskeleton is applied to the patient. In Figure 2b, the newly developed application creates an interactive interface between the patient and the exoskeleton that provides information about the current state of the finger and the exoskeleton and enables therapy via games. The data collected during this process can then be reviewed by the therapist or patient and used for therapy objectification and progress diagnosis (Figure 2c).

### 2.3. Application Details

Details of the application’s capabilities are shown in Figure 3.

The basis for this is the signal view in Figure 3a, which allows all relevant measurement signals of the finger to be viewed. These include the angular trajectories of MCP, PIP, and DIP joint (top view), the corresponding moment curves (middle view), and the measured force of the force sensor for validation. To validate the measurement signals and to obtain an intuitive image of the finger with the exoskeleton, a live view was created in Figure 3b. This allows one to view the finger in its current position based on the solution by Dickmann et al. An extension is the quasi-static solution of the equations for forces and moments in the exoskeleton. These are not only solved for the loads within the finger joints, but extended to the kinematic chain of the exoskeleton, allowing the loads occurring in the exoskeleton to be represented. Compression rods are dynamically animated and shown in red and tension rods in green, as shown in Figure 3b. Details of the kinematic solution and the quasi-static dynamic implementations can be found in the GitHub repository https://github.com/NikonPic/ExoApp (accessed on 20 May 2022).

To motivate the patient to participate in therapy, two different games were developed. In the first game, “Bubble Collector”, the pure movement of the fingers is intended to motivate and increase mobility. The live view of the exoskeleton serves as the basis for this game. The game engine generates random, slowly growing “soap bubbles” around the patient’s fingertip and displays them on the screen (see Figure 3c). As soon as the patient touches a soap bubble with the fingertip, it bursts and gives points indirectly proportional to its size. Thus, bursting smaller and harder-to-reach soap bubbles is rewarded with a higher score. The second game focuses on the patient’s dexterity, works specifically with the patient’s force feedback, and is shown in Figure 3d. The goal of the game is for the orange ball to survive as long as possible without colliding with a rectangle. The rectangles are randomly generated, move horizontally, and become faster over time. The movement of the orange ball is directly coupled with the force feedback of the exoskeleton and corresponds to the acceleration of the ball up or down.

A major advantage of the Festo-based kinematics is the adaptivity of the system, which can be applied to different hand sizes [6]. Only a few parameters are required from the patient, such as the respective lengths of the three phalanges of the index finger. The application provides a separate input mask for this purpose, which must be filled out once during a new registration. The patient-specific profile is then saved and can be edited at any time. In addition, the patient receives an overview of all his therapy sessions and the associated measurements. If the patient’s parameters change, the measurements are adjusted accordingly.

### 2.4. Application Games in the Context of Hand Rehabilitation

The goals of these developed games are to implement established concepts of classical physiotherapy and ergotherapy. The games offer the possibility to train the fine motor skills of the hand with and without resistance. The visualization and abstraction of the real patient’s finger onto a virtual model in the video game corresponds to a digital mirror therapy and is intended to utilize the therapeutic benefit of analog mirror therapy, which has been demonstrated in acute CRPS in controlled studies [19,20]. A similar approach is also taken by the “graded motor imagery” therapy concept in the classic CRPS treatment in which the imagination of movements is combined with mirror therapy and has shown very good therapeutic success [21,22]. In addition, the game-like and motivating character and the self-controlled exercise by the patient encourages a degree of movement at which certain pains are tolerated by the patient. This therapy concept is applied in the CRPS therapy under the concept of in vivo exposure (EXP), in which the patient intentionally accepts pain in order to achieve an improvement of the therapy success (no pain, no gain) and stands in contrast to conventional physiotherapy, in which pain is to be avoided during exercise (no gain with pain) [23].

### 2.5. Implementation of Clinical Studies

The study protocol was designed to treat patients with the exoskeleton during outpatient rehabilitation in addition to conventional physiotherapy. Six study sessions, each lasting 30 min, were conducted over 6 weeks. Every study session was carried out according to the defined protocol. Medical history was initially taken, including demographic data and the triggering event of CRPS. CRPS was classified using the Budapest diagnostic criteria [24], and the patient’s functional limitations in daily life, work, and sports were documented using the validated Quick-DASH score [25]. In addition, associated measurements were performed on the hands to assess other objectifiable parameters related to the progression of CRPS, which are shown in Figure 4.

Before and after each session, the (I) temperature of the hands with a thermal imaging camera (Seek Thermal CompactPRO), the (II) O2 saturation/circulation of the fingers with the pulse/heart-rate using a pulse oximeter (PULOX PO-200), and the (III) skin conductance with an impedance device (Mindfield eSense Skin Response) were measured and documented in a side-by-side comparison. The absolute values of the parameters can be used as indicators of the patient’s stress level and, as progression parameters, they can provide indications of the progression of the disease [26,27].

Subsequently, the range of motion, trajectory, and force of the hand parameters were measured via the applied exoskeleton with and without the motor. Then, the control games “Bubble Collector” and “Dodge Rectangles”, as well as the Leap Motion games, were performed over 20 min. Finally, the hand parameters and the accompanying measurements were performed again and documented. At the finalization of the experimental sessions after 6 weeks, the Budapest diagnostic criteria and the Quick-DASH scores were additionally collected a second time.

## 3. Results

### 3.1. Demographics

The presented exoskeleton was used to treat three CRPS patients over a period of 6 weeks according to the study protocol. These were two women and one man. In two patients, the CRPS was related to a fracture and a surgical procedure on the affected limb. In one patient, CRPS occurred as a result of overuse of the hand while writing a graduate thesis. The demographic data are summarized in Table 1. Each patient received at least 6 sessions of 30 min of treatment. During the first four weeks, one to two sessions were conducted per week, and the final treatment was given at the six-week follow-up. Patients were highly motivated and reported very pleasant session experiences. No complications or adverse events occurred during or following the trial sessions.

Patients were wearing the exoskeleton two times a week over a period of six weeks.

### 3.2. Accompanying Measurements

The accompanying measurements collected before and after each session (temperature of the hands, O2 saturation of the finger, heart rate, and skin conductance) were documented and evaluated. It was found that the individual parameters did not show any significant change in correlation to the individual treatment or the follow-up. A summary of the average values obtained is shown in Table 2.

### 3.3. Patient-Reported Outcome Measures (PROMS)

The evaluation of the recorded PROM and the subjective feelings of the patients showed improvements in the complaints and the functions of all three patients. The Budapest criterion (as a sign of the severity of CRPS) decreased in all patients. In addition, the evaluation of the DASH score showed improvement in all areas. The results of the patient follow-up study are shown in Table 3.

### 3.4. Exoskeleton Measurements

Further, we present the recordings of the joint and torque curves with the exoskeleton before and after the six weeks of therapy by the physical therapist. The results of this follow-up, as well as details about each patient wearing the exoskeleton, are shown in Figure 5.

The hands of the patients with the exoskeleton and the corresponding joints and moment curves are shown. The joint curves are displayed for the MCP, PIP, and DIP joints before and after the 6 weeks. The presented motion was generated by the linear motor of the exoskeleton, and the patient’s finger followed the flexion provided by the exoskeleton without active muscle actuation.

The angular measurements resulted from solving the exoskeleton’s kinematic system with the patient’s parameters. The force measurements also allowed the calculations of all corresponding torque curves of the finger joints. The torque curves are the quasi-static solutions of the loading equations [18]. They are shown under each of the joint curves. The curves shown and the corresponding shaded areas around them represent the mean and standard deviations over five measured trajectories. In patient 2, a complete trajectory could not be recorded prior to therapy because a predefined force limit of 6 N was exceeded.

## 4. Discussion

### 4.1. Applicability of the Exoskeleton System

The platform, consisting of the exoskeleton and mobile applications, was successfully applied to patients. The combination of the wearing comfort of the exoskeleton and the ease of use of the application enabled patients to use the system easily and intuitively. The storage of individual user profiles and the input masks of individual patient parameters allowed the measurements and life views to be optimized for the patient. The control system successfully responded to existing force limits, as shown in Figure 5 for patient 2 (green). Exceeding the force limit of 6 N stopped the movement and the finger was immediately unloaded.

### 4.2. Applicability of Gamification in Rehabilitation

The presented games could be successfully applied to the patient and formed the temporally largest part of the therapy session supported by the exoskeleton. The game “Bubble Collector” from Figure 3c has a similar differentiation to the real hand due to its similarity to the already established analog mirror therapy [19,20] and motivates an active movement of the hand. The game “Dodge Rectangles” from Figure 3d is controlled by the patient’s use of force. Thus, the patient can specifically control the maximum load during therapy within the limits of his/her pain tolerance and the patient is also motivated to allow higher forces in the interest of the success of the game.

### 4.3. Evaluation of Exoskeleton Measurements

The longitudinal study, which shows the progress of patients over a period of six weeks, contains remarkable observations. First, it can be observed that all three patients had positive therapeutic courses, as both QuickDASH [29] and Budapest [28] scores fell. This finding is also reflected in the torque curves from Figure 5. For patient 2, the trajectory could then be applied to the exoskeleton, as the stiffness of the finger was reduced and, therefore, the external force of the exoskeleton remained below 6 N during the trajectory. The joint torque curves (of the MCP joints) for patients 1 and 3 had significantly flatter values at the second 10 (point of maximum flexion) within the movement trajectory; this also applied (to a somewhat lesser extent) to the joint torque curves of PIP and DIP.

The joint angle trajectories of the three patients differed greatly from each other and also changed during therapy. The reason for the high variance between the patients was the adaptivity of the exoskeleton system. The externally applied force of the linear motor seeks the path of least resistance in the finger. Therefore, a patient with a relatively high joint moment in a finger joint has a rather low joint angle amplitude. The relationship between the joint angles and moment curves can be illustrated by patient 3 (orange) in Figure 5. Here, the joint moment curve of the MCP joint showed a significantly flatter course after the six weeks. As a result, the joint also exhibited less resistance to motion and the joint angle of patient 3 moved earlier in the trajectory than at the beginning of therapy. This also changed the joint angle curve of the PIP joint, as it no longer had to compensate for the reduced movement of the MCP joint and had a lower joint angle amplitude.

### 4.4. Comparison to Healthy Subjects

To put the measured patient data in context, the obtained results of the performed patient study are compared with the measurements from the healthy subject study of Dickmann et al. [18]. Both studies were conducted under the same conditions (the linear motor of the exoskeleton defined the fixed flexion motion). The comparative study of healthy subjects and patient tests are shown in Figure 6.

The top row shows the individual angular trajectories of the MCP, PIP, and DIP joints, and the bottom row displays the corresponding torque curves for all three subjects on the left and all three patients on the right.

When comparing the subject study by Dickmann et al. [18] and the presented patient study in Figure 6, several observations can be made. Because the exoskeleton was only allowed to actively apply an external force of 6 N to the patient, the trajectory could not initially be applied to patient 2 because the finger was too stiff at the beginning of the test. In addition, the torque curves of the healthy subjects and patients differed considerably. Patients consistently showed higher amplitudes in the joint torque curve. This can primarily be explained by the higher stiffness of the patient joints, which induced higher torques in the motion sequence.

### 4.5. Limitations

There are several limitations to our study. The first limitation involved the small number of patients and the short treatment phase with limited follow-up. This was due to the rare but severe clinical picture, in which the inclusion of patients in a clinical trial was significantly more difficult. In addition, regarding new medical device regulations (MDRs), the approval of a new medical product is considerably more difficult, so our study was only approved as a feasibility study on a small number of patients. However, the results are encouraging and should lead to an extension of the exoskeleton to the other three long fingers and enable us to extend our studies to more patients.

The second limitation was the insufficient reliability of the accompanying measurements. The accompanying measurements were performed to objectify the clinical course of CRPS using additional parameters. In the literature, temperature, O2 saturation, heart rate, and skin conductance are used to determine the influence of the sympathetic nervous system [26,27]. Skin conductance is often used to determine the phasic response (0.1–1 µS, event correlating value in the baseline difference) over the tonic level (2–20 µS, mean of absolute values over a long period of time) [30]. In our measurements, no correlation to the clinical course could be observed due to a very wide dispersion of the absolute values. The other parameters (temperature, O2 saturation, heart rate) were also influenced by environmental factors rather than by the test; we recommend the use of exoskeleton measurements to objectify the course of the disease for further studies. One solution for improved accompanying measurements would be to include more patients in the study and track data over a longer period of time in order to gather enough data for a more detailed analysis. Further, the upgrade to a more sophisticated skin conductivity measurement system is recommended.

## 5. Conclusions

The development of the mobile application and the integration and application on the patient could be successfully implemented and, thus, the exoskeleton of Dickmann et al. [18] could be extended. The operability with Bluetooth made it possible to flexibly and easily use the exoskeleton. The comparative study between the subject study of Dickmann et al. [18] and the conducted patient study showed clear differences in the torque curves for the patients, which could be attributed to the increased stiffness of the finger joints. The follow-up study showed a positive trend in QuickDASH and Budapest scores. This trend was also demonstrated with the exoskeleton, although it should be noted as a limitation that the number of patients was small. Furthermore, the applied system is not limited to CRPS by default, but can also be applied to different rehabilitation scenarios, such as strokes, and can be used as an extension of existing rehabilitation devices, such as the SPIDER system by Glowinski and Blazejewski [31]. The future goals of this project will be to extend the exoskeleton to the whole hand as well as extend the application.

## Figures and Tables

**Figure 1 sensors-22-04804-f001:**
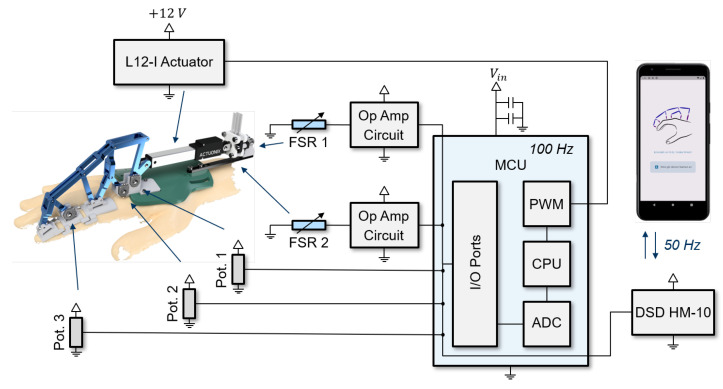
Overview of the electrical circuit of the extended exoskeleton by Dickmann et al. [18] and the corresponding use for the exoskeleton system.

**Figure 2 sensors-22-04804-f002:**
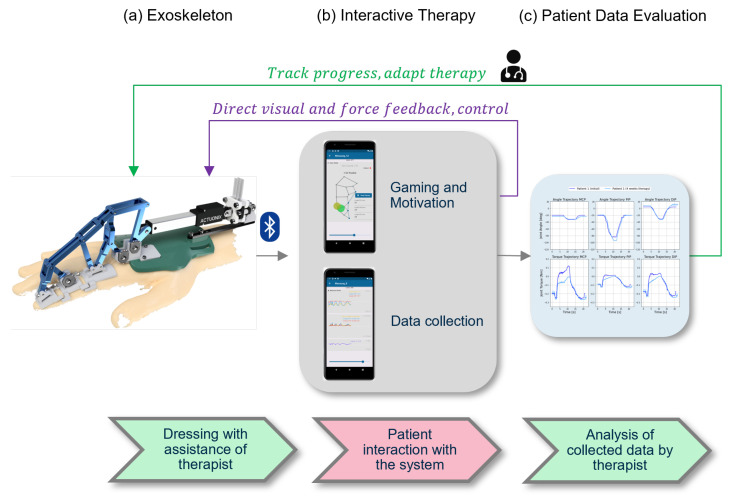
Overview of the workflow of the exoskeleton and the app. The exoskeleton from Dickmann et al. [18] in (**a**) is connected to the app via Bluetooth and enables games and data acquisition (**b**). The received data can be used for therapy or progress assessment (**c**).

**Figure 3 sensors-22-04804-f003:**
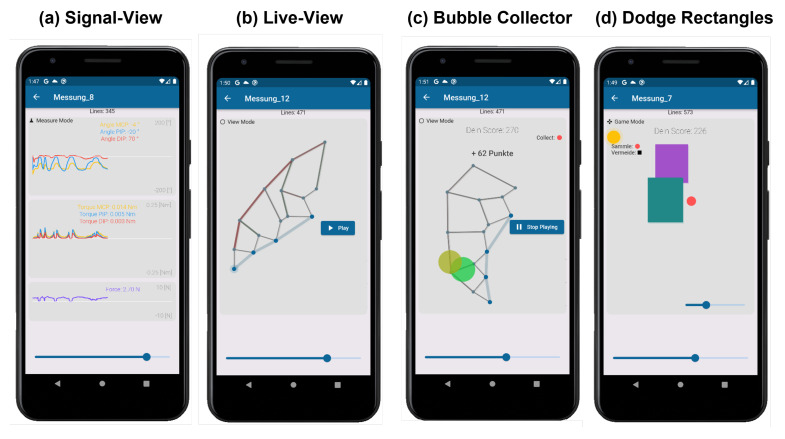
Application details. The general signals of the exoskeleton are displayed in (**a**) and the actual position of the exoskeleton in (**b**). To motivate the patient, the games “bubble collector” in (**c**) and “dodge rectangles” in (**d**) are displayed.

**Figure 4 sensors-22-04804-f004:**
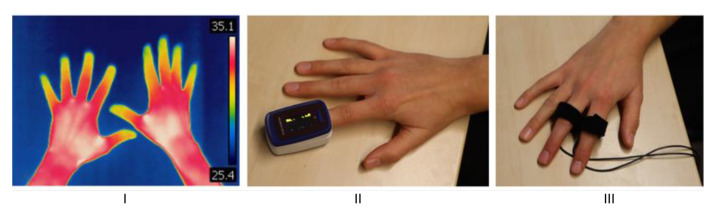
Used measurement tools for additional patient monitoring. Displayed are (**I**) the temperature by Seek Thermal CompactPRO, (**II**) pulse oximeter (PULOX PO-200), and (**III**) skin conductance.

**Figure 5 sensors-22-04804-f005:**
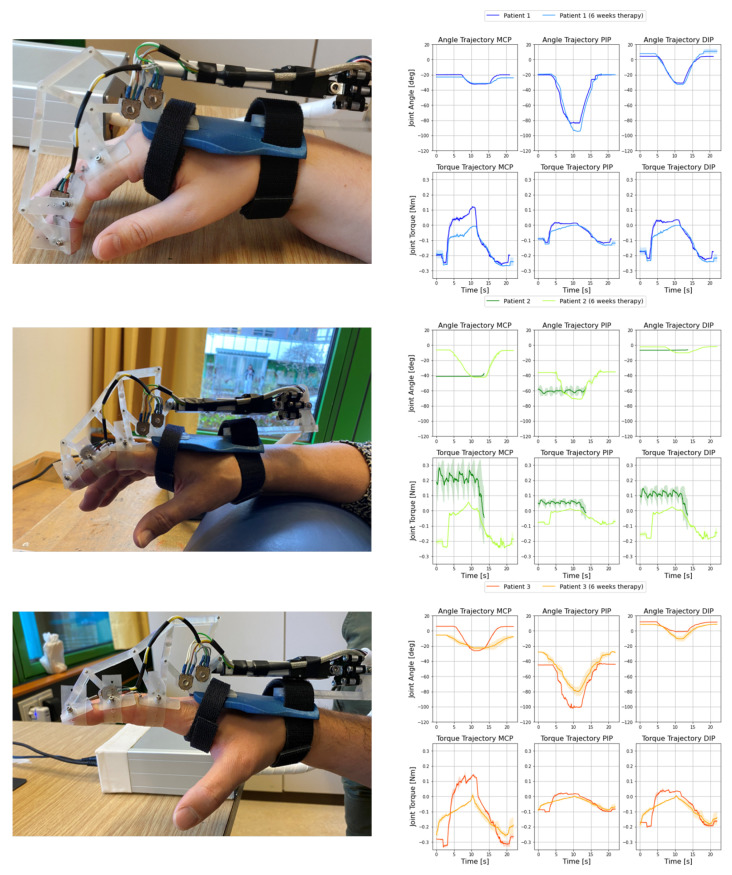
Comparative study of the exoskeleton of patient 1 (blue, **top**), patient 2 (green, **middle**), and patient 3 (orange, **bottom**) over a six-week period. On the left are the patient’s hands wearing the exoskeleton. On the right are the joint curves of the MCP, PIP, and DIP joints (top row) and the corresponding torque curves (bottom row). The measurements shown correspond to the mean (line) ± standard deviation (shaded) of five individual measurements.

**Figure 6 sensors-22-04804-f006:**
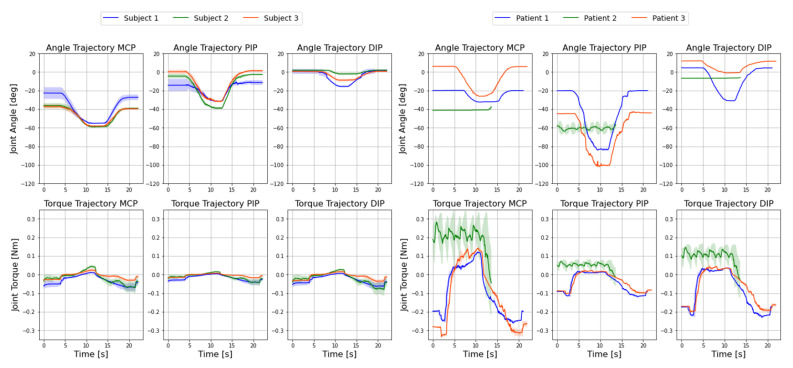
Comparative study between the three healthy subjects from Dickmann et al. [18] on the left and the three patients with CRPS on the right before therapy. The angle trajectories of MCP, PIP, and DIP joints are on the top row and the respective torque trajectories are on the bottom. The measurements shown correspond to the mean (line) ± standard deviations (shaded) of five individual measurements.

**Table 1 sensors-22-04804-t001:** Demographics of patients.

Patient	Age	Sex	BMI	Duration CRPS	Trigger CRPS
1	26	female	32.1 kg/m${}^{2}$	5 years	Hand overload when writing
2	68	female	28.3 kg/m${}^{2}$	7 months	Elbow fracture
3	44	male	21.3 kg/m${}^{2}$	8 months	Wrist fracture

**Table 2 sensors-22-04804-t002:** Overview of accompanying measurements of all patients in the six-week follow-up.

Patient	Temperature	O2-Saturation	Heart Rate	Skin Conductivity
1 pre	37 °C	90%	91 min	0.78 µS
1 post	37 °C	99%	82 min	0.34 µS
2 pre	38 °C	96%	58 min	0.38 µS
2 post	38 °C	96%	62 min	0.09 µS
3 pre	39 °C	97%	64 min	0.64 µS
3 post	36 °C	99%	60 min	0.33 µS

**Table 3 sensors-22-04804-t003:** Overview of patient follow-up according to QuickDASH and the Budapest score over a follow-up period of six weeks. In each case, a lower score corresponds to a better patient condition.

Score	Patient 1	Patient 2	Patient 3
Budapest Pre [28]	6/11	5/11	7/11
Budapest Post [28]	5/11	1/11	2/11
QuickDASH Pre [29]	50%	43%	40%
QuickDASH Post [29]	45%	38%	38%

## Data Availability

The data presented in this study are openly available and can be found in the GitHub repository https://github.com/NikonPic/ExoEval (accessed on 20 May 2022).

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
