# Peer review of "Development of an Exoskeleton Platform of the Finger for Objective Patient Monitoring in Rehabilitation"

_sensors, 2022, doi:10.3390/s22134804_

Round 1
Reviewer 1 Report
This paper deals with exoskeleton platform of the finger for objective patient monitoring in rehabilitationand is of great interest as one of the innovative treatment methods. It would be worthy of publication if the following points were corrected.
In the introduction, please detail the problems with conventional methods and how the methods developed by the authors in this study are expected to contribute to them.
The authors describe the limitations of their method in Section 4.5. Please describe any ideas you have for solving them.
It is well written. I consider it worthy of acceptance if the sections I commented on to the author can be corrected.
Author Response
Thank you for the overall positive feedback and the contributions on how to improve our research. We have tried to address your remarks in the following manner:
In the introduction, please detail the problems with conventional methods and how the methods developed by the authors in this study are expected to contribute to them.
The introduction has been revised to further highlight the problems with conventional therapy and highlight our contributions to therapy:
...
The conventional method of manual treatment is time consuming and often leads to unsatisfactory results, as therapy is often too short, requires an expert therapist and has high financial burdens [11 , 12]. Further, as patient data is not actively collected patient progress is only measured subjectively. In contrast, exoskeletons can provide accurate data to track the functional status of the patients hand, while not increasing the financial burden of the treatment over long periods or frequent applications. This enables patients to follow a daily therapy, usually not applicable in manual therapy and maintain therapy over a long period of time. 
...
The approach by Dickmann et al. [18] extends the approach by Conti et al. [6 ] and provides sufficient sensing for index finger actuation, motion tracking and force control. We will extend this approach to be suitable for clinical use, as a mobile application for control is developed and live tracking of patient data and easy applicability will be ensured. This will make it possible to simplify the use of the exoskeleton and provide the system as an extension of conventional therapy, objectifying the patient’s progress through the sensor data collected. In summary, we make the following contributions:
...
The authors describe the limitations of their method in Section 4.5. Please describe any ideas you have for solving them.
Thank you for pointing this out. We have extended the discussion to our limitations on the study in the following way:
There are several limitations to our study. First, the small number of patients and the short treatment phase with limited follow-up. This is due to the overall rare but severe clinical picture, in which the inclusion of patients in a clinical trial is significantly more difficult. In addition, the new MDRs (medical device regulations) the approval of a new medical product is considerably more difficult, so that our study was only approved as a feasibility study on a small number of patients. However, the good results are encouraging and should lead to an extension of the exoskeleton to the other 3 long fingers and the thumb for future applications and enable us to extend our studies with more patients.
Second, the insufficient reliability of the accompanying measurements. The accompanying measurements were performed to objectify the clinical course of CRPS using additional parameters. In the literature, temperature, O2 saturation, heart rate and skin conductance are used to determine the influence of the sympathetic nervous system [ 26, 27 ]. Skin conductance is often used to determine phasic response (0.1-1 μS, event correlating value in difference to baseline) over tonic level (2-20 μS, mean of absolute values over a long period of time) [30]. In our measurements, no correlation to the clinical course could 271
be observed due to a very wide dispersion of the absolute values. The other parameters (temperature, O2 saturation, heart rate) were also influenced by environmental factors rather than by the test, so that we recommend the use of exoskeleton measurements to objectify the course of the disease for further studies. A solution for improved accompanying measurements would be to include more patients in the study and track data over a longer period of time in order to gather enough data for a more detailed analysis. Further, the upgrade to a more sophisticated skin conductivity measurement system is recommended.
Reviewer 2 Report
Dear authors,
Your manuscript presents a very interesting mobile mechatronic system for finger rehabilitation.
The comparative results for the three patients presented in Fig. 5 evidenced the effectiveness of the proposed system.
From a mechanical point a view I propose to be continued the research in order to establish the grasping capacity of the rehabilitated finger (clamping force).
I propose to be accepted in present form.
Author Response
Dear Reviewer,
Thank you very much for revising the manuscript and the positive feedback. We will continue our research in the proposed manner and pay special attention to the grasping capacity of the rehabilitated finger.
Reviewer 3 Report
Dear Authors,
thank you very much for sending article titled: Development of an Exoskeleton Platform of the Finger for Objective Patient Monitoring in Rehabilitation. Looking at the title for the first time, I began reading the article with interest. The approach of the study appears very original. The contents of the manuscript are quite interesting by his methodology and through the tools used. Below are suggestions to authors:
- like authors wrote, the limitation is the small number of patients and the short treatment period. I suggest add in the title - a preliminary study
- in my opinion this method can be useful for people after stroke. For example, with additional exercises. Please look at the article titled: SPIDER as A Rehabilitation Tool for Patients with Neurological Disabilities: The Preliminary Research, jpm 2020. The authors used a special device to rehabilitate people after a stroke. The use of the additionally proposed method of rehabilitation could significantly contribute to improving the progress of rehabilitation. In order to improve the quality of the article, please quote the above article and write a few sentences about the possibilities of using the proposed method in the future (in summary).
Author Response
Dear Reviewer,
Thank you very much for taking the time to review our article. It is nice to hear that the article was read with interest.
like authors wrote, the limitation is the small number of patients and the short treatment period. I suggest add in the title - a preliminary study
Thank you for this hint. As our focus in this paper is on the development of the exoskeleton platform we have adapted the suggestion in the form that we added the proposed suggestion to the abstract:
We present the design of the exoskeleton, the mobile application with its game content and the results of the performed preliminary patient study.
In my opinion this method can be useful for people after stroke. For example, with additional exercises. Please look at the article titled: SPIDER as A Rehabilitation Tool for Patients with Neurological Disabilities: The Preliminary Research, jpm 2020. The authors used a special device to rehabilitate people after a stroke. The use of the additionally proposed method of rehabilitation could significantly contribute to improving the progress of rehabilitation. In order to improve the quality of the article, please quote the above article and write a few sentences about the possibilities of using the proposed method in the future (in summary).
Thank you for suggesting this interesting research. We have cited the article and discussed the applicability of the exoskeleton in the conclusion:
The development of the mobile application and the integration and application on the patient could be successfully implemented and thus the exoskeleton of Dickmann et al. [18] could be extended. The operability with Bluetooth made it possible to use the exoskeleton flexibly and easily. The comparative study between the subject study of Dickmann et al. [18 ] and the conducted patient study showed clear differences in the torque curves for the patients, which can be attributed to the increased stiffness of the finger joints. The follow-up study showed a positive trend in QuickDASH and Budapest scores. This trend was also demonstrated with the exoskeleton, although it should be noted as a limitation that the number of patients was small. Furthermore, the applied system is not by default limited to CRPS, but can also be applied to different rehabilitation scenarios, such as stroke, and can be used as an extension of existing rehabilitation devices such as the SPIDER system by Glowinski and Blazejewski [31]. The next goals of this project are to extend the exoskeleton to the whole hand and extend the application.